# An evaluation of the U.S. EPA's correction equation for PurpleAir Sensor data in smoke, dust, and wintertime urban pollution events

Daniel A. Jaffe[1,2], Colleen Miller[1,4], Katie Thompson[1], Brandon Finley[1], Manna Nelson[3], James Ouimette[5], Elisabeth Andrews[6,7]

[1]University of Washington Bothell, School of STEM, Bothell, WA 98011, USA

[2]University of Washington Seattle, Department of Atmospheric Sciences, Seattle, WA 98195, USA

[3]Seattle University, Department of Civil and Environmental Engineering, Seattle, WA 98122, USA

[4]Now at University of California-Irvine, Department of Chemistry, Irvine, CA 92697, USA

[5]Santa Rosa, CA 95409, USA

[6]NOAA Global Monitoring Laboratory, Boulder, CO 80305, USA

[7]Cooperative Institute for Research in Environmental Sciences, University of Colorado, Boulder, CO 80309, USA

Correspondence: Daniel A. Jaffe (djaffe@uw.edu)

## Abstract

PurpleAir Sensors (PASs) are low-cost tools to measure fine particulate matter (PM) concentrations and are now widely used, especially in regions with few regulatory monitors.  However, the raw PAS data have significant biases, so the sensors must be calibrated to generate accurate data.  The U.S. EPA recently developed a national correction equation and has integrated corrected PAS data onto its AirNow website.  This integration results in much better spatial coverage for $PM_{2.5}$  (particulate matter with diameters less than 2.5 µm) across the U.S. The goal of our study is to evaluate the EPA correction equation for three different types of aerosols: typical urban wintertime aerosol, smoke from biomass burning, and mineral dust.

We identified 50 individual pollution events, each having a peak hourly $PM_{2.5}$ concentration of at least 47 µg m$^{-3}$ and a minimum of 3 hours over 40 µg m$^{-3}$, and characterized the primary aerosol type as either typical urban, smoke, or long-range transported dust.  For each event, we paired a PAS sampling outside air with a nearby regulatory $PM_{2.5}$ monitor to evaluate the agreement. All 50 events show statistically significant correlations (R values between 0.71–1.00) between the hourly PAS and regulatory data, but with varying slopes.  We then corrected the PAS data using either the correction equation from Barkjohn et al. (2021) or a new equation that is now being used by the U.S. EPA for the AirNow Fire and Smoke Map (EPA, 2022b).  Both equations do a good job at correcting the data for smoke and typical pollution events, but with some differences.  Using the Barkjohn et al. (2021) equation, we find mean slopes of 1.00 and 0.99 for urban and smoke aerosol events, respectively, for the corrected data versus the

regulatory data. For heavy smoke events, we find a small change in the slope at very high $PM_{2.5}$ concentrations ($>600$ µg m$^{-3}$), suggesting a ~20 % underestimate in the corrected PAS data at these extremely high concentrations. Using the new EPA equation, we find slopes of 0.95 and 0.88 for urban and smoke events, respectively, indicating a

slight underestimate in $PM_{2.5}$ using this equation, especially for smoke events. For dust events, while the PAS and regulatory data still show significant correlations, the PAS data using either correction equation underestimates the true $PM_{2.5}$ by a factor of 5–6.

We also examined several years of co-located regulatory and PAS data from a site near Owens Lake, CA, which experiences high concentrations of $PM_{2.5}$ due to both smoke and locally emitted dust. For this site, we find similar

results as above; the PAS-corrected data are accurate in smoke but are too low by a factor of 5–6 in dust. Using these data, we also find that the ratios of PAS-measured $PM_{10}$ to $PM_1$ mass and 0.3 µm to 5 µm particle counts are significantly different for dust compared to smoke. Using this difference, we propose a modified correction equation that improves the PAS data for some dust events, but further work is needed to improve this algorithm.

**Introduction**

Low-cost air sensors are becoming a ubiquitous way for the general public to measure local air quality. There are now thousands of these sensors publicly reporting data in real time to the PurpleAir map (map.purpleair.com). As one example, there are more than 700 active PurpleAir sensors (PASs) in the Puget Sound region of Washington State (from Tacoma to Everett), compared to ~15 regulatory monitors in the same area. This provides an enormous increase in spatial information on $PM_{2.5}$ (particulate matter with diameters less than 2.5 µm). However, there are no

clear performance standards for accuracy or precision of low-cost sensors. Several studies have examined the performance of low-cost sensors, including the PAS (Singer and Delp, 2018; Li et al., 2020; Ardon-Dryer et al., 2020; Manibusan and Mainelis, 2020; Tryner et al., 2020). The PAS uses the Plantower PMS5003 laser sensor to count particles that scatter light in the optical range (particles greater than about 0.2 µm in diameter). Most outdoor PASs include two identical PMS5003 sensors that can be compared to enhance quality control. The PAS data can

be downloaded with two "conversion factors", CF=1 or CF=Atm. The two $PM_{2.5}$ values are nearly identical until 25 µg m$^{-3}$, but above this value the CF=1 will be greater. The exact algorithm used by the PAS to convert the Plantower data to mass concentration using either the CF=1 or CF=Atm factors has not been published (Ouimette et al., 2022).

Tryner et al. (2020) evaluated three low-cost PM sensors, including the PMS5003, by exposing them to five

different types of aerosols in the laboratory. They found that the ratios of PMS5003-reported to filter-derived $PM_{2.5}$ mass concentrations were inversely proportional to mass median diameter (MMD). Wood smoke had the smallest MMD, 0.42 µm; its PMS5003 $PM_{2.5}$ mass had a mean that was 2.5 times the filter-derived mass. Conversely, oil mist had the largest MMD at 2.9 µm; its PMS5003 $PM_{2.5}$ averaged only 0.23 times the filter-derived $PM_{2.5}$. These lab results are consistent with the physical-optical model developed for the PMS5003 by Ouimette et al. (2022).

The model predicted that the PMS5003 response decreases relative to an ideal nephelometer by about 70–90 % for particle diameters $\geq 1.0$ µm. This is a result of using a laser that is polarized, the angular truncation of the scattered light, and particle losses (e.g., due to aspiration) before reaching the laser. Their model predicted that the PMS5003

would underestimate $PM_{2.5}$ for dust particles by approximately 70–90 %, depending on the coarse particle size distribution.

The Plantower sensor reports PM mass concentrations in three bins ($PM_1$, $PM_{2.5}$, and $PM_{10}$) and particle counts in 6 size bins (>0.3, >0.5, >1, >2.5, >5, and >10 µm), presumably based on the pulse height of the scattered radiation, although the exact procedure is not documented by Plantower or PurpleAir.   The PAS also reports temperature, relative humidity (RH) and pressure.  A number of field and laboratory studies have found that the particle number size distributions reported by the PMS5003 are not correct. Several studies have reported that the PMS5003 tends to

create an invariant normalized size distribution, independent of the actual size distribution and concentration (Tryner et al., 2020; He et al., 2020; Kuula et al., 2020; Ouimette et al., 2022). However, the PMS5003 normalized size fractions above 1 µm increased by a factor of 2–5 in one high-$PM_{2.5}$ windblown dust episode observed at Keeler, California (Ouimette et al., 2022).  So, at present, there remains some ambiguity over how the PAS reported $PM_{2.5}$ mass concentrations and particle counts respond to different aerosol types.

Aerosol size distributions can vary considerably depending on the source type.  Previous studies have shown that the aerosol size distributions for smoke events are similar to the distributions in typical urban pollution events, with geometric mean diameter of around 0.2–0.3 µm (Kleeman et al., 1999; Laing et al., 2016). The mass ratio of $PM_{2.5}/PM_{10}$ for smoke, 0.55–0.75, is also similar to that for urban pollution (Xu et al., 2017). Dust events are known to have size distributions that are shifted towards larger particles, compared to typical urban and smoke aerosols.

Jiang et al. (2018) report an average $PM_{2.5}/PM_{10}$ ratio of 0.1 for dust events in China. Sugimoto et al. (2016), suggest a value of 0.35 for the $PM_{2.5}/PM_{10}$ ratio in dust, similar to the values reported by Tong et al. (2012).  In addition, aerosol particles from some cooking methods, such as barbeque, may also have a size distribution that is shifted to larger sizes (Kleeman et al., 1999; Song et al., 2018). If this is correct, then this may have implications for using PAS data to examine indoor air quality.

The South Coast Air Quality Management District (South Coast AQMD) has completed a rigorous evaluation of a variety of sensors, including the PAS (South Coast AQMD, 2015). This evaluation has shown that the PAS gave precise results, showed little response to temperature or humidity, and had relatively small variations between units. The U.S. Environmental Protection Agency (EPA) also provides information about these sensors via its Air Sensor Toolbox for citizen scientists, researchers, and developers portal (EPA, 2022a). All of these evaluations have

demonstrated that the raw PAS measurements are precise, but often biased high compared to regulatory $PM_{2.5}$ measurements in the United States.  Several groups have developed correction equations for the PAS measurements. The Lane Regional Air Protection Agency (LRAPA), the University of Utah, and the EPA have empirical corrections for $PM_{2.5}$ and these can be implemented directly on the PurpleAir website (PurpleAir, 2022).  Barkjohn et al. (2021) (hereafter referred to as Barkjohn 2021) conducted a comprehensive evaluation of PAS $PM_{2.5}$ data

against regulatory $PM_{2.5}$ data and developed a U.S.-wide correction equation starting from PAS raw data (CF=1) and using the  RH as measured by the PAS:

**Corrected PAS $PM_{2.5}$ = raw PAS $PM_{2.5}$ data (CF=1) * 0.52 – RH * 0.085 + 5.71**    (1)

The LRAPA and the Barkjohn corrections are in close agreement, whereas the University of Utah correction gives somewhat higher values. While the Barkjohn 2021 algorithim (equation 1) was initially used by the EPA, they have

recently developed a new correction algorithm that it is now being used for the national Fire and Smoke Map (Barkjohn et al., 2022).  This algorithm differs significantly from the earlier Barkjohn 2021 relationship in that it starts from the PAS data with CF=Atm and involves a more complex, 5-part piece-wise regression, with weighting to smooth the transitions between segments.  For our analysis, we will refer to the new algorithm as "new EPA". Note that the PAS data can be downloaded with either CF=1 or CF=Atm.   In the present analysis, we start from raw

data with CF=1 (for Barkjohn 2021) or CF=Atm (for the new EPA correction).  Figure S1 compares the raw CF=Atm data with the new EPA correction algorithm, and Figure S2 compares the Barkjohn 2021 and the new EPA correction for the data used in Part I of this analysis.

Because many PAS devices are now installed around the world, both outside and inside, they can experience a wide range of aerosol types.  Thus, it is essential to understand the accuracy and precision of the PAS for various aerosol

events, which could differ based on the particle size distribution or other aerosol characteristics.  In this study, we evaluated the Barkjohn 2021 correction  and the new EPA correction for 50 different aerosol pollution events, encompassing typical urban aerosols, as well as smoke and dust aerosols. Our goals are:

1.  Evaluate the accuracy of both correction equations for each aerosol type;
2.  Examine whether the correction changes at very high $PM_{2.5}$ levels (e.g., >250 µg m$^{-3}$);

3.  Identify whether the PAS data can provide an indication of the aerosol type and, if so, whether this information can be used to improve the correction algorithm.

Below we first describe data treatment and events and aerosol type identification.  Then we report on results comparing regulatory and PAS observations for different aerosol types for 50 short-term pollution events.  We also use a longer time series from a single site (Keeler, CA) that experiences frequent high dust and smoke pollution

episodes.  Our results demonstrate that the PAS sensors can give accurate $PM_{2.5}$ data in urban pollution and smoke, but more work is needed to develop an improved correction for dust aerosols.

**Methods**

**Part I: 50 paired sites**

For this analysis we identified 50 short term pollution events where the aerosols could be clearly characterized as

either typical urban, smoke or dust.  For these events, PAS data were downloaded for each sensor from the PurpleAir website (map.purpleair.com). The raw data (CF=1 and CF=Atm) were downloaded as hourly averages. Regulatory PM data for the nearest monitoring site were downloaded from the EPA Air Data website (https://www.epa.gov/outdoor-air-quality-data) or the AirNow-Tech website (airnowtech.org), except for data from the monitoring site at Portland Cully Helensview School in Portland, OR (AQS Id 410512011), which were

downloaded from the Oregon Department of Environmental Quality website (https://www.oregon.gov/deq/aq).

For each paired PAS-regulatory site, we identified an intense pollution event that had an hourly peak $PM_{2.5}$ value at the regulatory site >40 µg m$^{-3}$ for at least 3 hours.  We also required that there be a good correlation between the

regulatory and PAS data.  For the 50 events we analyzed, the correlation coefficients between the regulatory and PAS-corrected data ranged from 0.77 to 0.996.  For each pollution event, we identified the most likely source of

elevated aerosols: typical urban wintertime pollution, biomass burning smoke, or dust.  Table 1 summarizes the method used to characterize each pollution event. Table S1 provides details on each of the 50 individual events, including PAS site, regulatory site ID, event dates, and distance between the two sites.  The average distance between the PAS and regulatory sites was 5.4 km, with a range of 0-15 km.  As shown in Figure S3, there is no significant relationship between the correlation coefficient and distance between sites.

Typical urban pollution events were identified for the non-wildfire season (winter months) and with no evidence of smoke or dust.  The PM sources for those events reflect typical urban wintertime pollution (vehicles, power plants, industry, and residential wood combustion), and the $PM_{2.5}$ mass is dominated by particles with diameters in the range of 0.30–0.60 μm (Zhang et al., 2010; Herner et al., 2005).  The typical urban pollution events had peak hourly $PM_{2.5}$ values at the regulatory sites of 47–259 µg m$^{-3}$.

Smoke events were identified by elevated $PM_{2.5}$ during the summer fire season and confirmed using the Hazard Mapping System (HMS) Fire and Smoke Product (Rolph et al., 2009; Kaulfus et al., 2017).  The HMS product is derived from multiple satellite images and updated multiple times each day.  Details on the HMS product are given in the references above. The HMS imagery was obtained via the AirNow-Tech website.  The smoke events had the highest peak $PM_{2.5}$ values at the regulatory sites with peak hourly values of 60–713 µg m$^{-3}$.

Dust events were identified by examining large-scale spatial patterns of $PM_{2.5}$, media reports, and the measured $PM_{10}/PM_{2.5}$ ratios from regulatory sites, if available. In Part I of our analysis, all 6 dust events occurred during the well-known June 2020 Saharan dust cloud that was transported to the U.S. and impacted surface concentrations across the southern U.S. (Francis et al., 2020; Euphrasie-Clotilde et al., 2021; Pu and Jin, 2021).  This event brought huge amounts of dust to the southern U.S. and resulted in daily average $PM_{2.5}$ concentrations of 60–103 µg m$^{-3}$ at

many locations.  The six dust events included in our analysis had peak hourly $PM_{2.5}$ values at the regulatory sites of 52–72 µg m$^{-3}$.  Figure S4 shows the impact of this dust on $PM_{2.5}$ across the southeastern U.S.

In total, we identified 50 events as either typical urban, smoke, or dust, lasting from 24 to 528 hours. We verified that each had an operating PAS and a nearby regulatory monitoring site.  For typical urban pollution, 16 cases were identified, with the majority (13) located in California and the remainder in Utah. We identified 28 smoke cases,

with locations in Alaska, California, Idaho, Oregon, and Washington. Six dust cases were identified, with locations throughout the southeast U.S.  Of the 50 events identified, 17 have co-located regulatory $PM_{10}$ data (3 urban, 8 smoke, and 6 dust).  The event times were chosen to incorporate a few hours of low concentrations before and after the highest $PM_{2.5}$ concentrations to improve correlations.  The corrections on these low PAS values can sometimes yield negative values at high RH. If corrected PAS values were less than 2 µg m$^{-3}$, these values were excluded from

the calculation of correlation with the regulatory measurements.

**Data quality control**

The data were quality controlled and screened using four criteria:

1. Since most PASs contain two sensors, A and B, we compared mass concentrations from the two sensors and the data were used only if the values were within 30 %. Most values are much closer than this, with an average
difference of 10 % across all events considered (4.6 % for the Keeler, CA, PAS data).

2. The PAS raw A and B values were averaged and excluded if less than 1 µg m$^{-3}$.

3. The PAS values were corrected using the Barkjohn 2021 correction and included only if greater than 2 µg m$^{-3}$.

4. Regulatory PM$_{2.5}$ data must be greater than 1 µg m$^{-3}$ (there were a number of 0 and negative values in the EPA's PM$_{2.5}$ data records).

In total, these steps removed approximately 10% of the available data. After screening, the PAS data were corrected using the Barkjohn et al (2021) algorithm and the new EPA correction algorithms. We evaluated both sets of corrected PAS data using the same linear relationship using standard linear regression:

$$\textbf{Regulatory data = Slope * PAS data (corrected) + Intercept} \qquad \textbf{(2)}$$

We also compared the slopes with reduced major axis regression (RMA) and found essentially no difference in the
results. Generally, the intercepts were small (a few µg m$^{-3}$), so we can interpret the slopes as giving the overall indication of agreement between the two datasets. A slope near 1 with a zero intercept would indicate no bias. A slope <1 indicates that the corrected PAS data are biased high compared to the regulatory data, a slope >1 indicates the corrected PAS data are biased low compared to the regulatory data.

**Part II: Keeler, CA, site**

To further understand the nature of the PAS response to dust aerosol, we also used data from Keeler, CA, near Owens Lake. Owens Lake is a dry lakebed due to diversion of its primary water source, the Owens River, to Los Angeles. As a result, the dry lakebed is one of the largest sources of dust in North America (Cahill et al., 1996; Gillette et al., 1997), and the region experiences many significant dust events each year. With increasing drought, it appears that the dust flux from Owens Lake is increasing (Borlina and Rennó, 2017). We use regulatory PM$_{2.5}$ and
PM$_{10}$ data from February 2019–May 2022 from a site in Keeler CA and a nearby PAS site. Both the regulatory and PAS instruments are operated and maintained by the Great Basin Unified Air Pollution Control District (GBUAPCD, Chris Howard, personal communication, Dec. 2022) and the regulatory data were obtained from their data archive (https://www.gbuapcd.org/). The regulatory PM$_{2.5}$ was measured using a Thermo Fischer model 1400a TEOM with a model 8500C conditioning system. For PM$_{10}$, a Thermo Fischer model 1400a TEOM was used from
February 2019 through September 2021 and a model 1405 TEOM was used from October 2021 through May 2022. Other information about the site is given in Table S2. The Keeler PAS and regulatory sensor inlets are within four meters of each other. For the Keeler PAS data, as in Part I, we use the mean of channels A and B, which have a mean difference of 4.6 %. For the Keeler analysis, we did not specifically identify events. Instead, we consider only hours where the Keeler regulatory PM$_{2.5}$ >25 µg m$^{-3}$, which provides 1366 hours of data, spanning a 3.3 year
period. We also restrict the analysis of the Keeler data to hours where regulatory PM$_{10}$ exceeds PM$_{2.5}$ by at least 0.5 µg m$^{-3}$ and where simultaneous regulatory and PAS data are available. This yields 1257 hours of data with mean PM$_{2.5}$ and PM$_{10}$ concentrations from the regulatory monitors of 59 and 118 µg m$^{-3}$, respectively.

## Results

### Part I: Event analysis

Figure 1 shows time series plots of two example events (# 44 and # 45).  The top plot in Figure 1 shows PAS and regulatory data during a major smoke event in Washington State during July–August 2021.  The regulatory $PM_{2.5}$ exceeded 200 µg m$^{-3}$ at this site.  This figure demonstrates that the Barkjohn 2021 correction yields excellent bias correction of the data.  The new EPA also improves the fit, compared to the raw data, but appears to yield a positive bias at the highest concentrations (200–250 µg m$^{-3}$).  The bottom plot in Figure 1 shows data from a dust period in

June 2020 (event # 45).  In contrast to the smoke event, the raw PA data is much lower than the regulatory data and both correction algorithms significantly under-predict the regulatory values and there is essentially no difference between the two correction schemes.  While there is still a good correlation between the regulatory and PAS data (R value of 0.82), the slope is 6.76, indicating that both correction equations are significantly under-estimating the true concentrations by a factor of 6 or more.  Table S3 shows the results for each of the 50  individual events. Table 2

summarizes the relationship and correlation slopes between the corrected PAS data and the regulatory measurements for the 50 events and for the three different aerosol types.  The results are consistent with Figure 1 in that urban pollution and smoke events are reasonably corrected by either the Barkjohn 2021 or new EPA algorithms, whereas dust events are not.    There are some differences between the two correction equations, which we discuss below.

Table 3 and Table 4 summarize the results by aerosol type and includes all hourly data for each identified aerosol

type.  Table 3 uses the Barkjohn 2021 correction, whereas Table 4 shows results using the new EPA correction.  For urban, smoke, and dust aerosols, the slope of regulatory $PM_{2.5}$ versus the PAS-corrected data with the Barkjohn 2021 algorithm were 1.02, 1.08,  and 4.98, respectively, using all hourly data of each type (Table 3).  Using the new EPA correction, the slopes were 0.95, 0.81, and 4.99, respectively (Table 4).  These slopes indicate that both correction algorithms yield excellent bias correction for typical urban and smoke events, but they exacerbate the

large negative (low) bias for dust events.  Using either correction on the PAS data during dust events gives values that are low by a factor of 5–6.

Tables 2, 3 and 4 suggest that the new EPA algorithm has slightly lower slopes, especially for the smoke events.  For example, the mean slope for smoke events (shown in Table 2) is 0.99 for the Barkjohn 2021 correction, vs 0.88 for the new EPA correction.   Similarly, using hourly data for smoke influenced periods the slopes are 1.08 using the

Barkjohn 2021 correction (Table 3) vs 0.81 using the new EPA correction (Table 4).   We also want to examine whether there is evidence that the PAS data respond differently at very high PM concentrations. Figure 2 shows the mean bias using the hourly data with both correction algorithms versus the regulatory $PM_{2.5}$.  This plot includes only data during the urban and smoke events.  The bias is strongly negative using the Barkjohn 2021 correction at very high $PM_{2.5}$, greater than about 300 µg m$^{-3}$.  At medium high $PM_{2.5}$ concentrations, such as 150–300 µg m$^{-3}$, the new

EPA correction shows a positive bias, which is consistent with the results shown in Figure 1a and Tables 2-4.  Thus we conclude that the new EPA correction improves the bias at very high concentrations (>300 µg m$^{-3}$),  but introduces a modest bias at moderately high pollution levels (150–300 µg m$^{-3}$), compared to the Barkjohn 2021 algorithm.

We show above that the raw PAS data, along with both corrections, are substantially under-reporting $PM_{2.5}$
concentrations during dust events. The next question is whether the PAS data can give some information about dust
events (i.e., the presence/absence of dust), despite significant issues with the reported size distribution (Ouimette et
al., 2022). To address this question, we calculated the slopes of the $PM_1$ to $PM_{10}$ mass ratios and the 0.3 µm to 5 µm
particle counts ratio, using the PAS data for each event. The results are reported by event type in Table 2. The
results show that the PAS reports a greater fraction of coarse mass and proportionally more larger particles,
compared to the 0.3 µm particles, in the dust aerosols, compared to urban or smoke aerosols. Both the $PM_1/PM_{10}$
mass ratio and the 0.3 µm to 5 µm particle counts ratio increases in the order dust < smoke < urban. These
differences are statistically significant ($P<0.05$) for urban versus dust using a two sample, two tailed t-test and
assuming unequal variance. These relationships will be explored further below in Part II of this analysis.

We also looked at the coarse aerosol fraction (CAF) for these events using both the regulatory and PAS data. We
define the CAF as:

$$CAF = (PM_{10} - PM_{2.5}) / PM_{10} \quad (3)$$

Out of the 50 events considered, 17 have both regulatory $PM_{2.5}$ and $PM_{10}$. Figure 3 shows the CAF, calculated using
both the regulatory data and the PAS raw data for all hours for the 17 events with both $PM_{2.5}$ and $PM_{10}$ data. For the
PAS data, we use the raw values (CF=1) for $PM_{2.5}$ and $PM_{10}$, since there are no known correction algorithms for the
$PM_{10}$ data. Several things are apparent in Figure 3. First, the CAF values using the regulatory data are much higher
than CAF values obtained from the PAS data. Nonetheless, both the regulatory and PAS data show the expected
pattern of higher CAF in dust compared to the other aerosol types. In addition, the number of data points is much
higher for the PAS, due to the relative sparsity of regulatory $PM_{10}$ data. We note that these relationships change
very little if the PAS data are restricted to the same times as the regulatory data.

**Part II: Keeler, CA, analysis**

In Part II we use the multi-year dataset from the Owens Lake/Keeler, CA, site. The hourly data cover a period of a
little more than 3 years (February 2019–May 2022). We focus exclusively on hours with regulatory $PM_{2.5} >2.5$ µg
$m^{-3}$, which yields 1257 hours, after our quality control described above. Table S2 has more details on both the
regulatory and PAS sites in Keeler, CA.

Figure 4 shows a histogram of the CAF based on the regulatory data. There is a clear bimodal distribution,
indicating two very different aerosol types during these pollution events. Given that Keeler is ca 150 km from the
urban areas of Bakersfield and Fresno, CA, this aerosol is likely either dust generated from Owens Lake or smoke
from the many California wildfires during 2019–2022. For the points with CAF<0.5 (n=1013 hours), the vast
majority (99 %) occurred in August–October 2020 or August–September 2021, both times when large fires were
burning in central California and influencing air quality across the region. Thus, it is reasonable to conclude that
those hours with CAF<0.5 are predominantly wildfire smoke (1013 hours), and those with CAF>0.7 (200 hours) are
predominantly dust. In contrast to the smoke data, the dust data tend to occur in the winter and spring periods.

There are a relatively smaller number of points (44 hours) with $0.5 < \text{CAF} < 0.7$ and these appear to have a mixed character, as shown below.

Table 5 and Figure 5 show results grouped by the CAF calculated using the regulatory data. Tables shows that for all values of CAF below 0.5, there are similar ratios of $PM_1/PM_{10}$ and 0.3 µm/0.5 µm counts. For this group, the PAS $PM_{2.5}$ with the Barkjohn 2021 correction shows a good fit to the regulatory $PM_{2.5}$. For the values with CAF>0.7, there is similar consistency in the PAS-measured ratios ($PM_1/PM_{10}$ and 0.3 µm/5 µm counts), but for this group the PAS Barkjohn 2021 correction significantly underestimates the regulatory concentrations. For the group

with CAF between 0.5 and 0.7, the aerosol has a mixed character, likely including both smoke and dust.

Figure 5 shows a plot of the regulatory $PM_{2.5}$ versus PAS $PM_{2.5}$ with the Barkjohn 2021 correction, sorted by these three groups (CAF<0.5, 0.5<CAF<0.7, and CAF>0.7). For the smoke aerosols, the PAS with the Barkjohn 2021 correction shows a slope of 0.99 and an $R^2$ of 0.92, whereas for the dust aerosols, the slope is 5.6, similar to the slopes shown in Table 2 (5.5) and Table 3 (5.0). Thus, we conclude that for dust aerosols the Barkjohn corrected

PAS values show a 5–6x underestimate of the $PM_{2.5}$ regulatory values. The mixed aerosols show behavior that is more difficult to characterize, with some showing more similarity to dust and others to smoke.

Figure S5 and S6, show the 0.3 µm/5 µm counts and the ratios of $PM_1/PM_{10}$, as measured by the PAS versus the CAF, and Table 5 shows a summary of the data segregated by CAF. Both ratios of $PM_1/PM_{10}$ and the 0.3 µm/5 µm counts show clear differentiation for the low CAF aerosols compared to the high CAF aerosols. So these unitless

ratios provide a tool that can identify dust aerosols, so that a separate correction can be applied. We explored both the $PM_1$ to $PM_{10}$ mass concentrations and the ratio of 0.3 µm to 5µm counts as tools to identify $PM_{2.5}$ aerosol that is dominated by dust. Figure S5 and Table 5 show that using a ratio of the 0.3 µm to 5µm counts of somewhere between 150–250 will provide the best separation of dust and mixed aerosols. By examination of various plots of regulatory $PM_{2.5}$ versus corrected PAS $PM_{2.5}$ for the Keeler, CA, data, we found an optimum value of 190. The

value of 5.6 comes from the slope of the dust aerosols in Figure 5. So, this leads to a new correction equation that depends on PAS-measured values:

**If PAS 0.3 µm / 5 µm > 190, use Barkjohn 2021 correction**

**If PAS 0.3 µm / 5 µm < 190, use Barkjohn 2021 correction * 5.6         (4)**

In Eq. (4), we use the Barkjohn 2021 correction, but in practice there is little difference in the results regardless of

whether this or the new EPA correction is used. Figure 6 shows a plot of the Keeler, CA, regulatory $PM_{2.5}$ versus PAS $PM_{2.5}$ with Eq. (4) applied. There is very little change to the smoke data as most of these points have PAS-measured 0.3 µm/5 µm counts >190. For the dust aerosols, the majority of the data points are now much closer to the regulatory values. The mean bias for the points with CAF >0.7 is now 1.3 µg m$^{-3}$ compared with 51.4 µg m$^{-3}$ for the dust data using the Barkjohn 2021 correction. Figures S7 and S8 show how the choice of 0.3 µm/5 µm ratio

impacts the analysis. Using a higher threshold in Eq. (4) results in identifying some points (smoke) with corrected $PM_{2.5}$ values that are substantially too high. Using a lower threshold in Eq. (4) results in missing some dust points and, for those points, generating PAS-corrected $PM_{2.5}$ values that are too low. While using a value of 190 in Eq. (4)

does miss a small number of dust points, it appears to be the best balance in finding and correcting the dust data points for this location.  Finally, Figure S9 shows regulatory $PM_{2.5}$ versus PAS $PM_{2.5}$ with the new EPA correction separated by CAF.  The results are nearly identical to Figure 5, showing that both the Barkjohn 2021 and new EPA correction algorithms have similar behavior with dust aerosols.

Equation (4) was developed based on data from one site (Keeler) that has strong dust and smoke occurrence and with the sensors in close proximity (30m).  We apply Eq. (4) to the 50 events from different sites identified in Part I and find a wider range of results.  Table S3 summarizes the results for each event. Out of the 6 dust events, 4 show moderate improvements with slopes of 0.46–0.72.  However, for some smoke events (e.g., 38, 39, and 40), the slopes are dramatically lower, in the range of 0.17–0.26, which indicates that the PAS-corrected with the dust algorithm (equation 4) are overestimating the regulatory data by a large amount.  This occurs due to the fact that during these smoke events some hours have a ratio of the 0.3 µm to 5 µm counts of >190 and thus get multiplied by 5.6.  So, while the new dust algorithm does appear to improve PAS-corrected data in dust events at a single controlled site that is operated by an air quality agency, it does not provide a useful correction for the bulk of publicly operated sensors.  Nonetheless, the fact that the PAS data indicate changes in the observed ratios of $PM_1/PM_{10}$ and the 0.3 µm/ 5 µm counts during mineral dust events indicates that the PAS data do provide some useful information on dust and that more work to identify a suitable correction algorithm for dust is warranted.

**Conclusion**

PASs are now ubiquitous around the world and far outnumber the more accurate, regulatory-grade instruments for $PM_{2.5}$.  These low-cost sensor data are proving to be highly valuable for a variety of analyses, but especially for improving our understanding of the spatial distribution of $PM_{2.5}$.  However, to use these data, it is essential to understand the measurements.  Using the Barkjohn 2021 and new EPA correction algorithm for PAS data, we find that the sensors give reasonably accurate results for $PM_{2.5}$ for typical urban wintertime pollution and smoke events, but give concentrations that are a factor 5–6 too low for dust events.  The Barkjohn 2021 algorithm yields a negative bias at very high $PM_{2.5}$ concentrations ($>300$ µg m$^{-3}$), whereas the new EPA algorithm yields a positive bias at moderate $PM_{2.5}$ concentrations (150–300 µg m$^{-3}$).  Both algorithms underestimate $PM_{2.5}$ during dust events by a factor of 5–6. Using the PAS ratios of $PM_{10}$ to $PM_1$ mass concentrations and 0.3 µm to 5 µm counts, we find that there are significant differences in these ratios for smoke and dust at a site with frequent incursions of both aerosol types.  Using this result, we propose a new PAS correction algorithm that significantly improves the correction for dust aerosols and does not change the results for smoke aerosols, but only at this one site.  Applying this equation to a broader array of sites, we find significant problems with the proposed dust algorithm—it improves PAS $PM_{2.5}$ estimates in some dust cases but worsens PAS $PM_{2.5}$ estimates for some smoke events.  Nonetheless, our analysis demonstrates that it may be possible to develop an improved PAS correction algorithm that could identify dust and provide a better estimate of the $PM_{2.5}$ concentrations when dust is present.

**Data availability.**  All data used in this analysis are publicly available.  Most regulatory data were obtained from the EPA Air Data site (https://www.epa.gov/outdoor-air-quality-data) and the AirNow-Tech site

(https://www.airnowtech.org/). Data for the Keeler, CA, site were from the Great Basin Unified Air Pollution
Control District (https://www.gbuapcd.org/). Data for the Cully Helensview School in Portland, OR, were
downloaded from the Oregon Department of Environmental Quality website (https://www.oregon.gov/deq/aq).
PurpleAir data were from the PurpleAir site (http://map.purpleair.com)

**Author contributions.** DJ designed the study, developed the analysis protocols, and wrote the manuscript. CM, KT, and MN conducted data analysis. BF, JO, and EA reviewed the manuscript and provided comments on the
analysis.

**Competing interests.** The authors have no competing interests to declare.

### Acknowledgements

We wish to acknowledge and thank the many individuals that have made their PAS data freely available for scientific analysis. Partial support for this work came from the UW Bothell SRCP Seed Grant Program. MN was
supported by an internship from the Confederated Tribes of the Colville Reservation, which was funded by an EPA Environmental Justice grant. EA was supported by the NOAA Cooperative Agreement with CIRES, NA17OAR4320101.

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

**Tables/Figures**

**Table 1.  Methodology for identification of pollution events for 50 cases in Part I.**

| Event | Method of identification | $PM_{2.5}/PM_{10}$ (if available) |
|---|---|---|
| Typical urban | One-hour regulatory $PM_{2.5}$ measurements exceeded 47 µg m$^{-3}$ during non-wildfire season with no known presence of smoke or dust. | >0.5 |
| Smoke | One-hour regulatory $PM_{2.5}$ measurements exceeded 47 µg m$^{-3}$ in the presence of smoke as indicated on the NOAA Hazard Mapping System-Fire and Smoke Product. | >0.5 |
| Dust | One-hour regulatory $PM_{2.5}$ measurements exceeded 47 µg m$^{-3}$ during a known dust event. | <0.5 |

**Table 2. Peak regulatory PM$_{2.5}$, mean slope and R$^2$ results from analysis of regulatory and PAS data, with Barkjohn 2021 correction and new EPA correction, for 50 individual pollution events (Part I dataset). N gives number of events of each type, SD is standard deviation. R$^2$ is the mean value for all events of that type. Also shown are the average slopes by aerosol type for the PM$_1$ versus PM$_{10}$ and 0.3 versus 5 μm counts regressions, both of which are unitless.**

|  | Average peak regulatory PM$_{2.5}$ (μg m$^{-3}$) | Average slope (R$^2$) using Barkjohn 2021 correction | Average slope (R$^2$) using new EPA correction | Average slope of raw PAS PM$_1$ versus PM$_{10}$ mass concentrations | Average slope of raw PAS 0.3 μm versus 5 μm counts |
|---|---|---|---|---|---|
| **Urban-avg (N=16)** | 85.15 | 1.00 (0.88) | 0.95 (0.88) | 0.56 | 727 |
| **SD** | 56.69 | 0.11 | 0.15 | 0.10 | 426 |
| **Smoke-avg (N=28)** | 280.32 | 0.99 (0.93) | 0.88 (0.92) | 0.44 | 402 |
| **SD** | 226.28 | 0.18 | 0.13 | 0.10 | 265 |
| **Dust-avg (N=6)** | 59.76 | 5.54 (0.85) | 5.53 (0.85) | 0.29 | 133 |
| **SD** | 7.91 | 1.13 | 1.10 | 0.08 | 77 |



**Table 3. Relationship between hourly regulatory PM$_{2.5}$ and corrected PAS PM$_{2.5}$ with Barkjohn 2021 algorithm. Data are included for all simultaneous measurements for the 50 identified events in Part I. (N gives number of hours of data of each type.) .**

| | Mean regulatory PM$_{2.5}$ (µg m$^{-3}$) | Mean corrected PAS PM$_{2.5}$ (µg m$^{-3}$) | Slope for regulatory versus PAS w/Barkjohn 2021 correction (R$^2$) | Intercept (µg m$^{-3}$) | RMSE* (µg m$^{-3}$) | Mean bias (µg m$^{-3}$) |
|---|---|---|---|---|---|---|
| **Urban (N=966)** | 33.9 | 28.7 | 1.02 (0.793) | 4.60 | 10.9 | -5.2 |
| **Smoke (N=6536)** | 66.4 | 66.0 | 1.08 (0.866) | -4.68 | 36.0 | -0.4 |
| **Dust (N=240)** | 30.5 | 6.4 | 4.98 (0.661) | -1.09 | 27.9 | -24.1 |

*Root mean squared error


**Table 4. Relationship between hourly regulatory PM$_{2.5}$ and corrected PAS PM$_{2.5}$ with new EPA algorithm. Data are included for all simultaneous measurements for the 50 identified events in Part I. (N gives number of hours of data of each type.)**

| | Mean regulatory PM$_{2.5}$ (µg m$^{-3}$) | Mean corrected PAS PM$_{2.5}$ (µg m$^{-3}$) | Slope for regulatory versus PAS w/new EPA correction (R$^2$) | Intercept (µg m$^{-3}$) | RMSE* (µg m$^{-3}$) | Mean bias (µg m$^{-3}$) |
|---|---|---|---|---|---|---|
| **Urban (N=966)** | 33.9 | 30.3 | 0.950 (0.744) | 4.90 | 11.1 | -3.6 |
| **Smoke (N=6536)** | 66.4 | 77.3 | 0.807 (0.858) | 5.56 | 43.2 | 11.0 |
| **Dust (N=240)** | 30.5 | 6.4 | 4.99 (0.664) | -1.22 | 27.9 | -24.1 |

*Root mean squared error



**Table 5. Mean regulatory (reg) PM$_{2.5}$, PAS PM$_{2.5}$ (with Barkjohn 2021 correction and with proposed dust correction), ratio of PAS PM$_1$/PM$_{10}$ raw concentrations (CF=1), and ratio of PAS 0.3 to 5 µm counts by coarse aerosol fraction (CAF) bins. The CAF bins are centered on the indicated value.**

| CAF bin midpoint | N (hrs) | Regulatory PM$_{2.5}$ (µg m$^{-3}$) | PAS PM$_{2.5}$ w/Barkjohn 2021 correction (µg m$^{-3}$) | PAS PM$_{2.5}$ w/dust correction (µg m$^{-3}$) | Mean ratio of PAS PM$_1$/PM$_{10}$ | Mean ratio of PAS 0.3 to 5 µm counts |
|---|---|---|---|---|---|---|
| 0.05 | 260 | 89.5 | 91.4 | 91.4 | 0.55 | 730 |
| 0.15 | 334 | 59.4 | 61.5 | 61.5 | 0.55 | 697 |
| 0.25 | 231 | 41.6 | 43.4 | 43.4 | 0.56 | 723 |
| 0.35 | 131 | 37.6 | 38.5 | 38.5 | 0.54 | 623 |
| 0.45 | 57 | 36.9 | 37.3 | 37.3 | 0.54 | 611 |
| 0.55 | 14 | 40.6 | 25.1 | 33.0 | 0.44 | 474 |
| 0.65 | 30 | 52.5 | 16.0 | 45.7 | 0.33 | 249 |
| 0.75 | 104 | 68.4 | 13.5 | 63.8 | 0.25 | 151 |
| 0.85 | 86 | 59.3 | 11.2 | 60.7 | 0.20 | 105 |
| 0.95 | 10 | 57.2 | 12.4 | 66.1 | 0.21 | 111 |


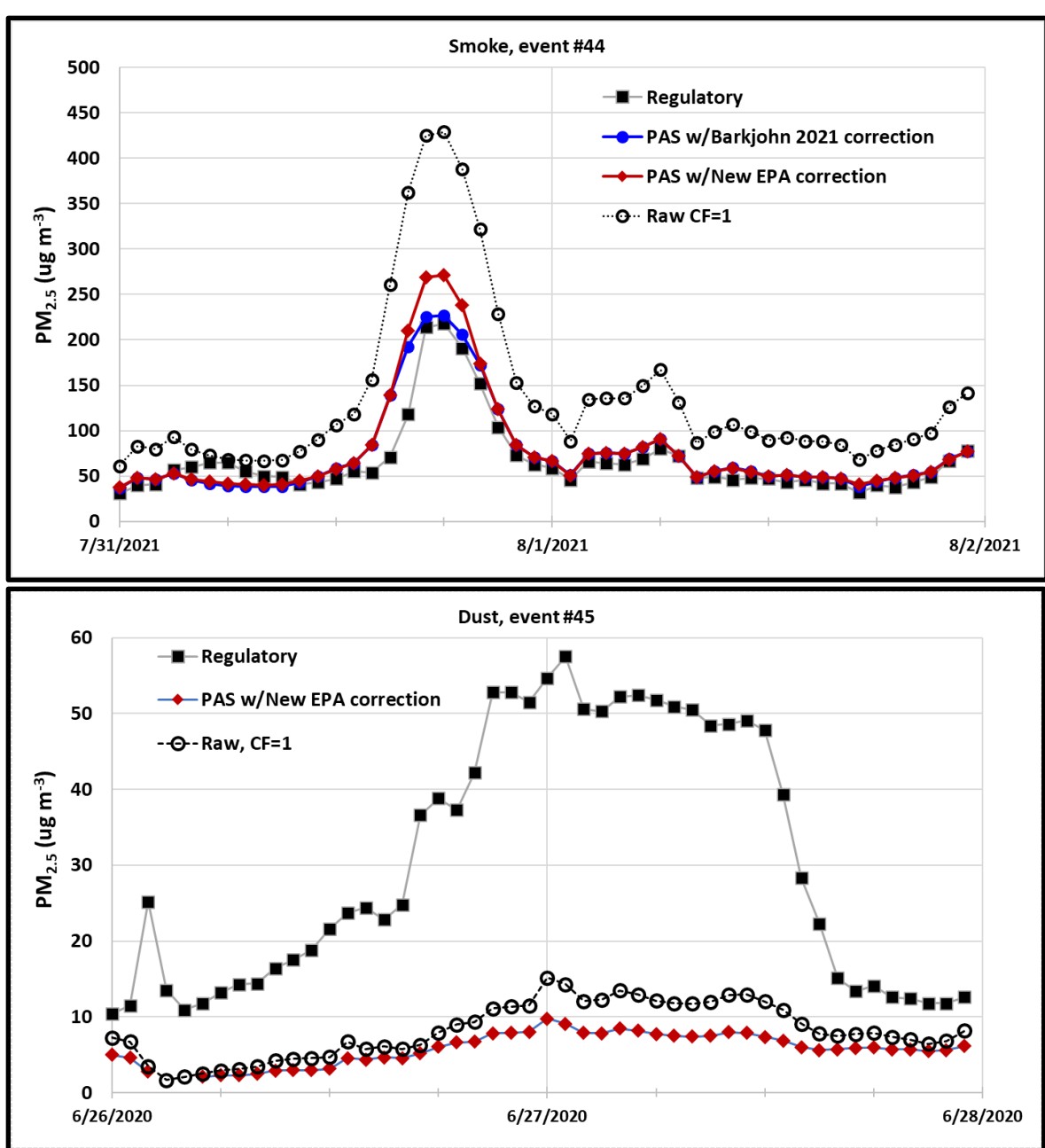

**Figure 1: Time series of hourly regulatory and PAS data PM$_{2.5}$ (raw and corrected) for two events, # 44 (smoke, top) and # 45 (dust, bottom). Time is in UTC. Note that for the dust event (# 45), the two correction schemes give identical results. Details on the sites used for these figures are given in Tables S1 and S3. For event 44, the slopes (using Eq. (2)) comparing the Barkjohn 2021 and new EPA corrections schemes are 0.81 and 0.70, respectively. For event 45, the slope using the Barkjohn 2021 correction scheme is 6.76.**



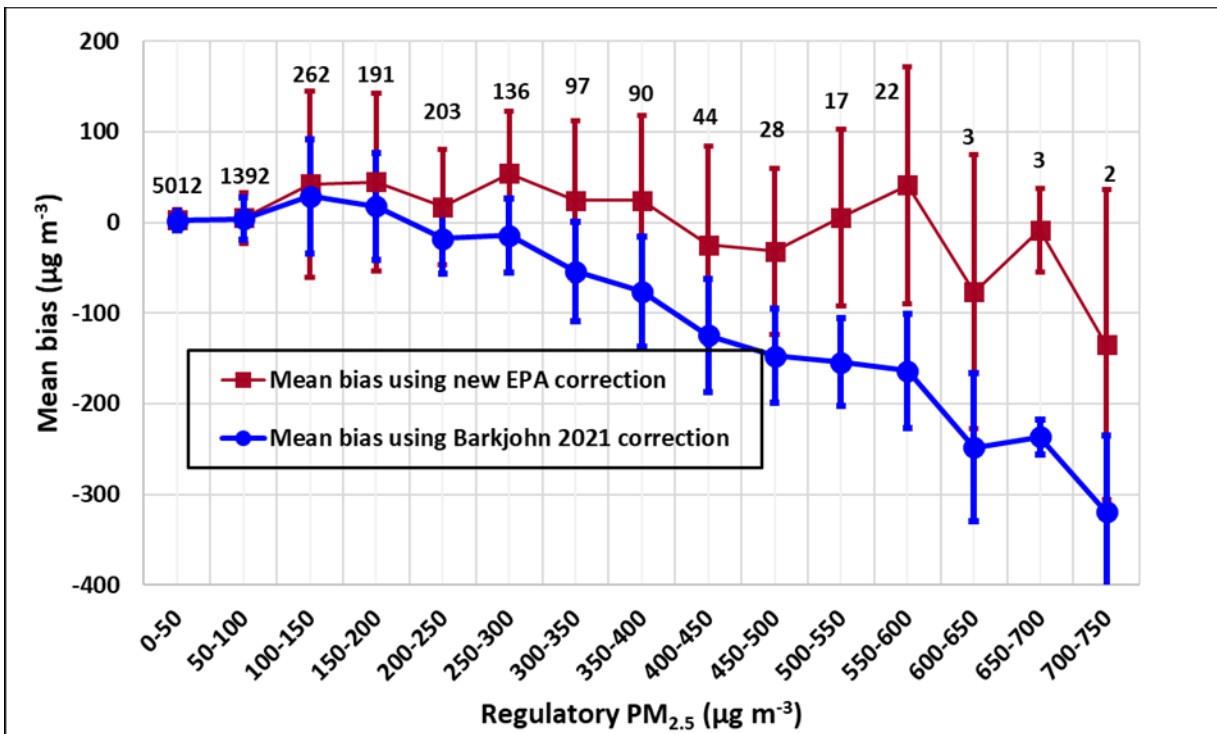

**Figure 2: Comparison of mean bias (corrected PAS-regulatory) using the hourly data for smoke and urban pollution events in Part I using the Barkjohn 2021 and new EPA correction schemes. Data are binned by regulatory PM$_{2.5}$ in 50 µg m$^{-3}$ bins, as shown on the X axis. The values above the red points are the number of hourly datapoints in each bin, which is the same for both the Barkjohn 2021 and new EPA corrected values. The error bars show one standard deviation within that bin.**

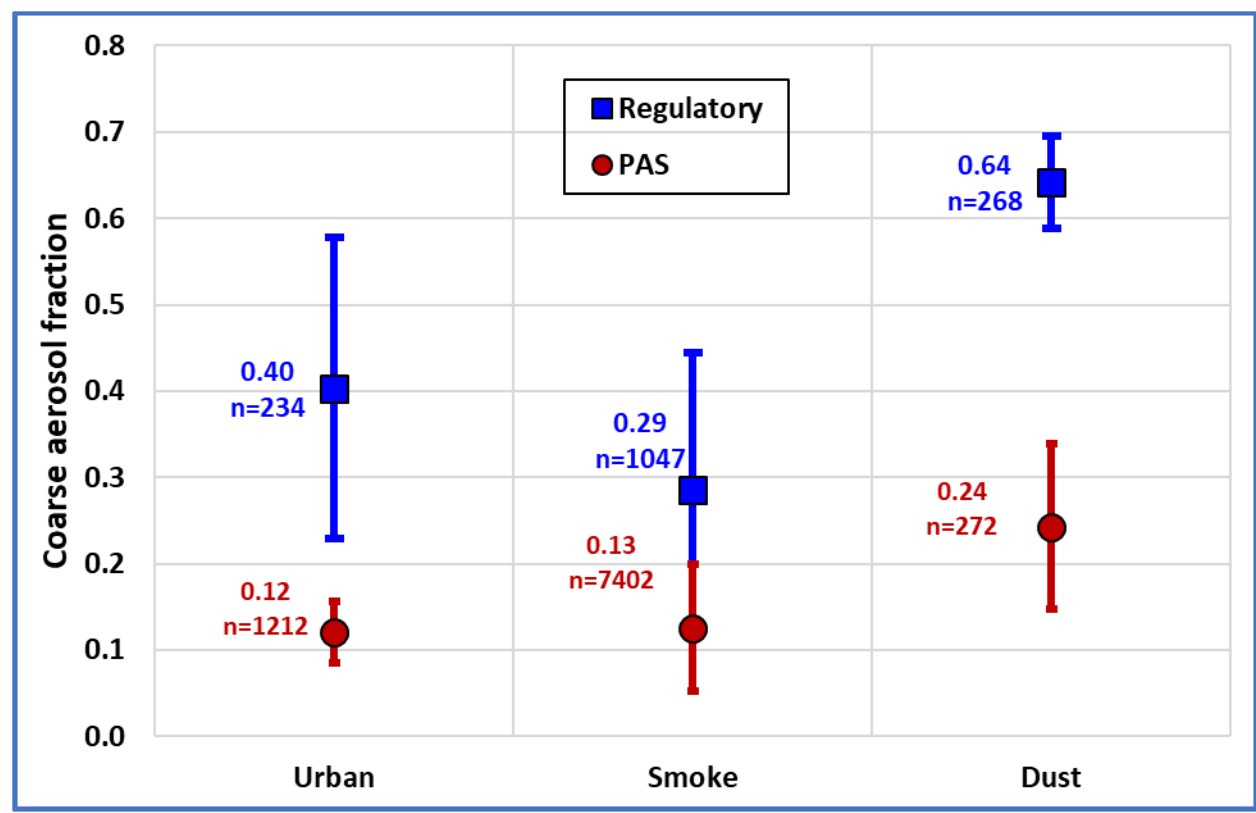

**Figure 3:** Mean coarse aerosol fraction (CAF) (Eq. (3)) calculated using the regulatory data and the PAS raw (CF=1) data for 17 events from the Part I dataset that had both PM$_{2.5}$ and PM$_{10}$. The values near each point give the mean and number of data points (hours) in each bin.


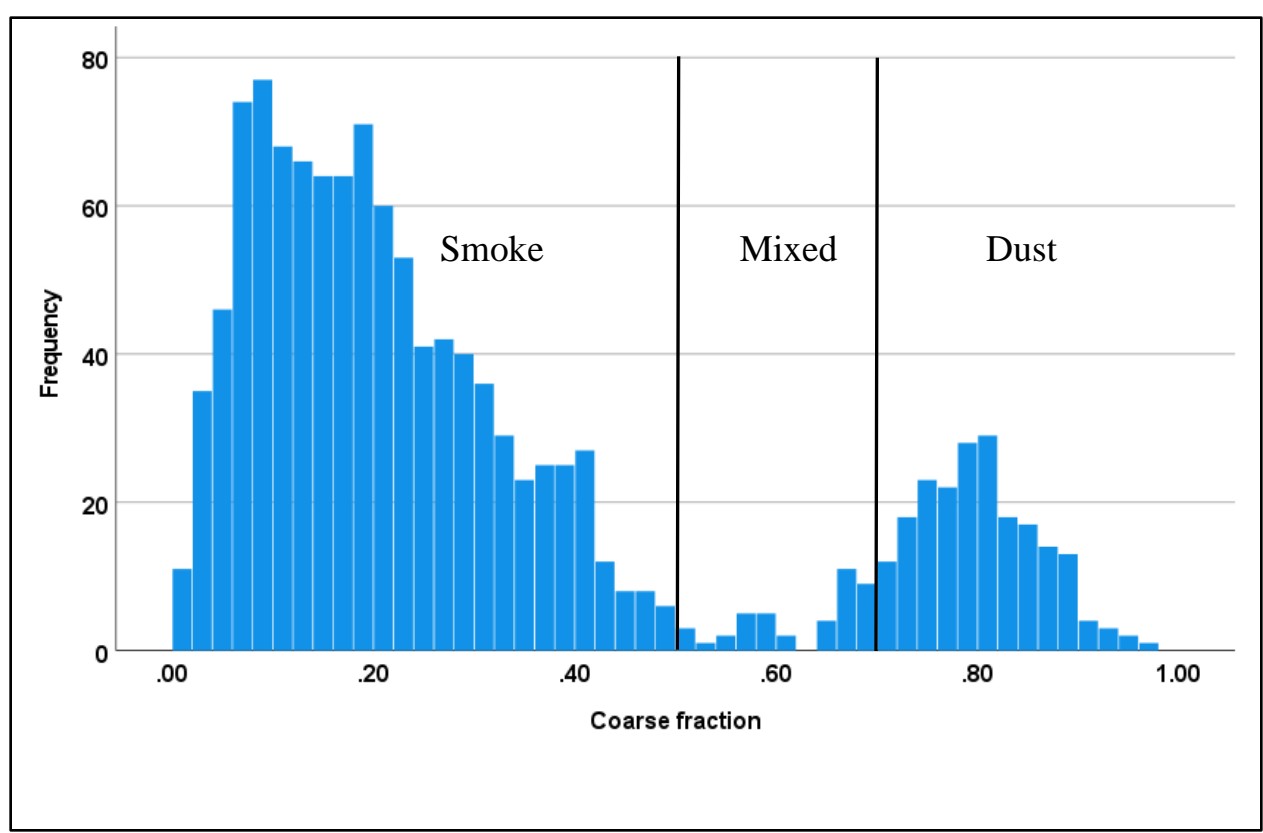

**Figure 4.  Histogram of coarse aerosol fraction (CAF) at Keeler, CA, using regulatory PM$_{2.5}$ and PM$_{10}$ data for hours with PM$_{2.5}$ > 25 µg m$^{-3}$.  We assume that the aerosol is primarily smoke when CAF<0.5, mixed when CAF is between 0.5 and 0.7, and dust for times with CAF>0.7.**



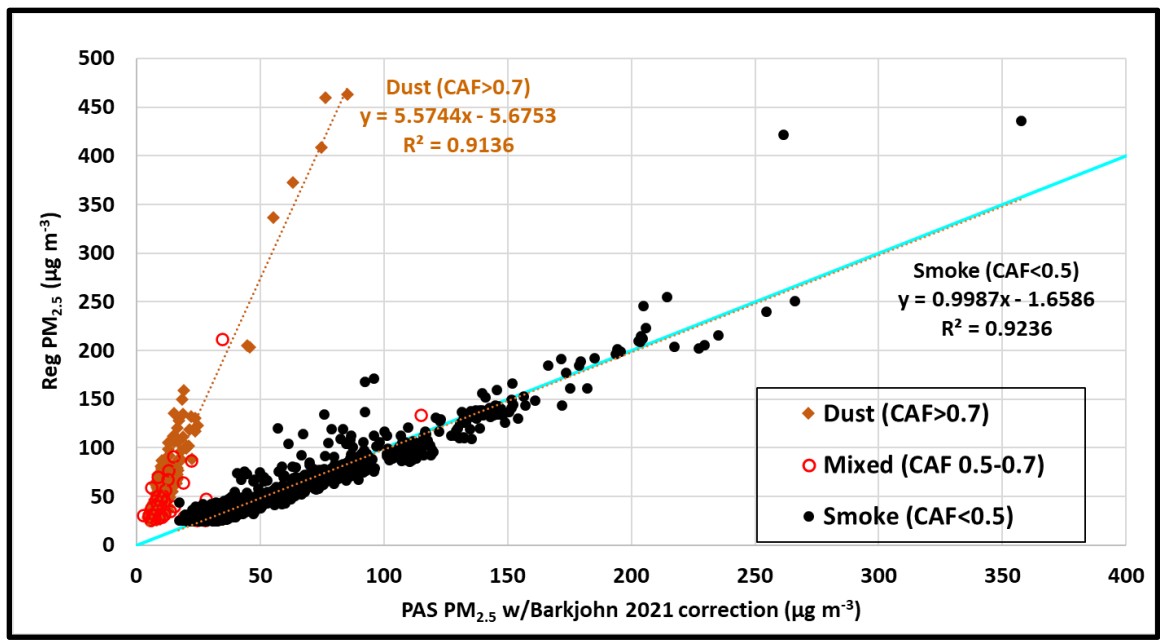

**Figure 5. Regulatory PM$_{2.5}$ versus PAS PM$_{2.5}$ with Barkjohn 2021 correction at Keeler, CA, for hours with regulatory PM$_{2.5}$ > 25 µg m$^{-3}$. The data are separated by the coarse aerosol fraction (CAF), as measured by the regulatory data. Linear regression relationships are shown with dotted lines and the light blue line shows a 1:1 relationship.**



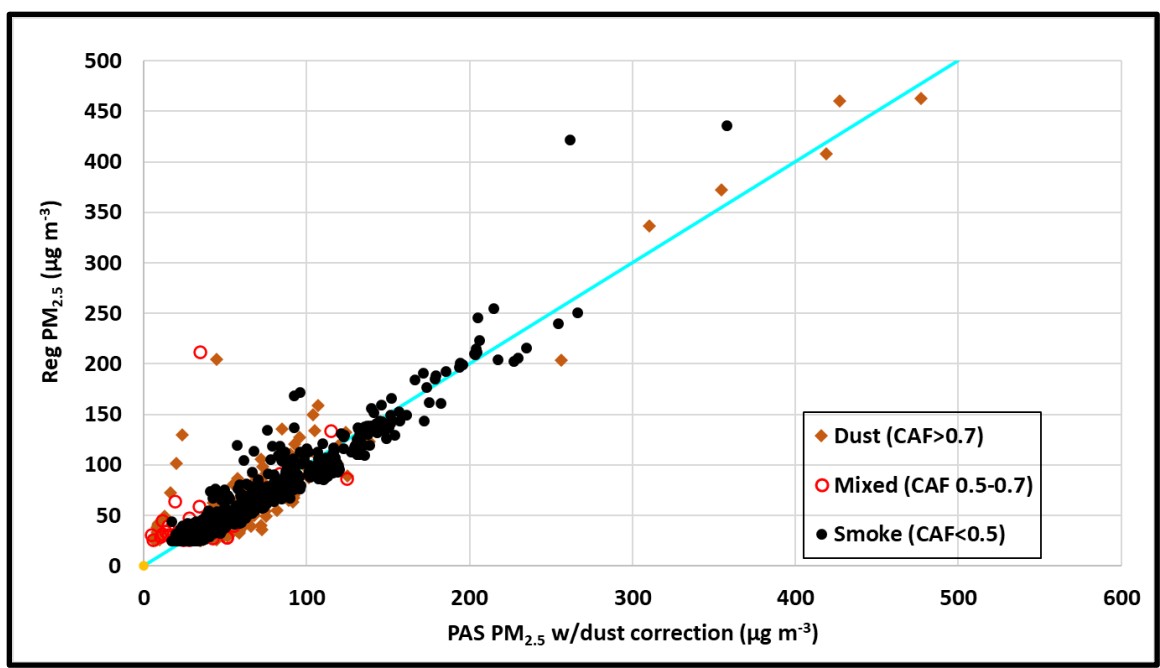

**Figure 6. Regulatory PM$_{2.5}$ versus PAS PM$_{2.5}$ with dust correction (equation 4) at Keeler, CA, for hours with regulatory PM$_{2.5}$ > 25 μg m$^{-3}$. The data are separated by the coarse aerosol fraction (CAF), as measured by the regulatory data. The light blue line shows a 1:1 relationship.**
