# Peer review of "An evaluation of the U.S. EPA's correction equation for PurpleAir Sensor data in smoke, dust, and wintertime urban pollution events"

_Atmospheric Measurement Techniques, 2022_

## Referee Comment (RC1)

**Review for An evaluation of the U.S. EPA's correction equation for Purple Air Sensor data in smoke, dust and wintertime urban pollution events by Jaffe et al.**

Overall, the paper is interesting, and nice findings are presented in it. However, I do think additional work is needed to improve the paper writing mainly to clarify some unclear points and improve the writing. I think additional work is needed on the result part, for example, expanding and showing additional figures for more locations/examples. The quality of the writing for section 2 of the dust seems different from the other parts, I highly encourage the author to improve that part, expand the explanations and also expand the discussions which seem very low. While the work on the dust is very important, I think the author should distinguish between the two dust events used as part of part 1 and part 2. When you define dust events as part of the 50 events reported it is important to note these are long-transfer dust and not local ones (as in part 1) these two types might be different (different particle sizes and concentrations).

There is a threshold for dust based on the  $PM_{2.5}/PM_{10}$  ratio (Line 75), which is a value lower than 0.35 (Sugimoto et al 2016), there had been some work in the US that show these values for dust event (e.g., Tong et al., 2012; Ardon-Dryer et al., 2022). Why you did not use a threshold for the dust part? Can you show how it's different (maybe less efficient) than what you developed?

**General Comments**

Line 122- would be nice to see more information about the pairs, as location on a map, mainly for non-US readers who are not familiar with many of the locations you mentioned. For Supplement Table S1 please add how many parallel times (h) were used for this comparison. Also, for the AQS names, it would be efficient to separate the EPA sensors based on their name as state-county-sensor as appear on the EPA website (e.g., 04-013-4002, 06-027-1003) would be easier for readied to allocate the sensors. Please the ID of the PAS sensors. Is the location for PAS or EPA sensor make sure both are provided in the table. In Line 130, you indicated there were no differences in the pair even when the distance was up to 15 km this is a very important aspect and more information on these pairs should be provided as many of the previous studies used pairs if the distance was less than 1 km. for Lines 132- 148 it would be good to mention the period used for these analyses, as it is unclear what time point was used for each event (e.g., indicate month and years data were retrieved for)

The information in lines 149 to 154 should be incorporated in the text when you talk about each one of the events and also a map showing these sensors identified is needed.

Line 161- can you clarify what you mean by values were excluded if less than 1  $\mu$ g m-3?

The sentence in line 163 seems not in place and would need more explanation and examples, how high does RH? show examples of how much data had to be removed due to this fact.

Line 156-167 - Would be important to add information based on the data quality sections on how much data had to be removed due to each screening criteria (is case = event?). would have been efficient to show more examples not just one for dust and one for smoke. Show additional figures, and provide information in the text about the different examples so we will understand how Tables 2-3 were created.

Line 175- map needed for a non-American reader.

Line 180 – nearby- how close, map, and distance will be helpful for the reader.

Line 192 would be good to know when PAS and location are used or in this example, information could be added to table S1 so the reader will know what sensor was used for what event.

Line 211-216- the entire paragraph is unclear including fig 2 how were these calculated made, how many events were used to correlate the number of cases, did only a period of the event (e.g., smoke) was used, or sometimes data of period before or after as shown in Fig 1 were used. The R2 values are not significant so can you clarify what statistically significant are you talking about?

Line 230- what 17 events? You said there are 50 cases, can you indicate maybe in table S1 which one had both PM sizes that were used for this analysis? Why not use the known threshold which is commonly used ( $PM_{2.5}/PM_{10}$ ) for this comparison? Why showing uncorrected PAS values? What does the corrected PAS look like? Can you repeat this part for corrected PAS values? Can you show a fig of PAS before and after corrections?

Line 246- did you confirm the events were dust based on observation of meteorology conditions, and satellite?  $PM_{2.5}/PM_{10}$  ratios? Please add more information, and add dates of these events so the reader may be able to understand what you have done for this part. I think this is one of the most interesting parts of your paper but it seems you put the least amount of effort into it.

Line 266-267 what are the units for 150-250 and 190 are these nm or  $\mu g m^{-3}$ ?

Lines 285-289- show us these findings of the use of the new eq on the dust event of part one, this is a very important finding from this paper, and it seems the author barely provides information here. Also, would be nice to see exploring these sections a bit more, and show us similar to fig 1 what the dust events look like before and after using both corrections( time serious) was this one dust event or several, and when did they occur, how long they lasted?

**Minor comments**

Line 54 and 55 - PM2.5 the 2.5 should be lowercase ( $PM_{2.5}$ ), there might be more of these that I missed

Table 2-P value based on which statistical test?

Fig 4- units of the axis are missing

**Reference used**

Sugimoto, N., Shimizu, A., Matsui, I., and Nishikawa, M.: A method for estimating the fraction of mineral dust in particulate matter using PM2:5-to-PM10 ratios, Particulogy, 28, 114–120, https://doi.org/10.1016/j.partic.2015.09.005, 2016.

Tong, D. Q., Dan, M., Wang, T., and Lee, P.: Long-term dust climatology in the western United States reconstructed from routine aerosol ground monitoring, Atmos. Chem. Phys., 12, 5189–5205, https://doi.org/10.5194/acp-12-5189-2012, 2012

Ardon-Dryer, K., Kelley, M.C., Xueting, X., and Dryer, Y.: The Aerosol Research Observation Station (AEROS), Atmos. Meas. Tech., https://doi.org/10.5194/amt-2021-270, 2022.

---

## Author Comment (AC1)

**Original review comments in black. Author response in red.**

**Reviewer #1:**

Overall, the paper is interesting, and nice findings are presented in it. However, I do think additional work is needed to improve the paper writing mainly to clarify some unclear points and improve the writing. I think additional work is needed on the result part, for example, expanding and showing additional figures for more locations/examples. The quality of the writing for section 2 of the dust seems different from the other parts, I highly encourage the author to improve that part, expand the explanations and also expand the discussions which seem very low.

Thank you for your positive comments. We will edit this section to be as clear as possible.

While the work on the dust is very important, I think the author should distinguish between the two dust events used as part of part 1 and part 2. When you define dust events as part of the 50 events reported it is important to note these are long-transfer dust and not local ones (as in part 1) these two types might be different (different particle sizes and concentrations).

Yes, this is an important point and will be noted in the final text.

There is a threshold for dust based on the PM2.5/PM10 ratio (Line 75), which is a value lower than 0.35 (Sugimoto et al 2016), there had been some work in the US that show these values for dust event (e.g., Tong et al., 2012; Ardon-Dryer et al., 2022). Why you did not use a threshold for the dust part? Can you show how it's different (maybe less efficient) than what you developed?

There is no single threshold value that works in all situations. It will depend on the dust source, transport distance, etc. For our analysis, the data in Figure 4 show a clear bimodal distribution with a split near a course fraction of 0.5-0.65. Converting this to fine fraction (PM2.5/PM10), yields a separation value of 0.35-0.5, which is similar to the values mentioned above. Given that there are very few points in this range, the exact value chosen to separate the data set has little impact on the analysis. This discussion will be expanded and the references above added.

General Comments Line 122- would be nice to see more information about the pairs, as location on a map, mainly for non-US readers who are not familiar with many of the locations you mentioned. For Supplement Table S1 please add how many parallel times (h) were used for this comparison. Also, for the AQS names, it would be efficient to separate the EPA sensors based on their name as state-countysensor as appear on the EPA website (e.g., 04-013-4002, 06-027-1003) would be easier for readied to allocate the sensors. Please the ID of the PAS sensors. Is the location for PAS or EPA sensor make sure both are provided in the table.

EPA uses both styles to indicate an AQS site #. In the interest of space, we choose to keep the AQS # as they are. This gives an unambiguous identification of the site. We also note that states and EPA sometimes use different site names for identical locations with the same AQS numbers. For this reason, the AQS # is the best way to identify each of the regulatory sites. The lat and long for these sites can be easily found on the EPA or state websites. There is more ambiguity in the PA site names. Identical PA sites names are sometimes used to identify different PA sensors. This is why its important to give both the PA site name and coordinates.

In Line 130, you indicated there were no differences in the pair even when the distance was up to 15 km this is a very important aspect and more information on these pairs should be provided as many of the previous studies used pairs if the distance was less than 1 km.

The plot below shows the correlation coefficient as a function of distance between the site pairs. As noted already we see no significant reduction in correlation for these pollution events for our site pairs as a function of distance. But keep in mind that this is only for the short term pollution event that we identified. We choose not to include this plot in the manuscript, but it can be easily generated from the information in Tables S1 and S2.

[Figure]

for Lines 132- 148 it would be good to mention the period used for these analyses, as it is unclear what time point was used for each event (e.g., indicate month and years data were retrieved for)

This information is present in Table S1 (Start time, End time) and a reference to this table will be included in this paragraph for better clarity.

The information in lines 149 to 154 should be incorporated in the text when you talk about each one of the events and also a map showing these sensors identified is needed.

There are 50 different site pairs, plus the Keeler CA analysis. Its not clear what spatial information the reviewer is requesting. We disagree that a map of these 100 sites will provide any useful information that is relevant to the analysis.

Line 161- can you clarify what you mean by values were excluded if less than 1 µg m-3?

We have added the following text. "The event times were chosen to incorporate a few hours before the start of the high $PM_{2.5}$ concentrations to improve correlations." In some cases, very low $PM_{2.5}$ values were included in this, as reported by the PAS. The corrections on these low PAS values can yield negative values at high RH. If raw PAS values were less than 1 µg m-3 these values were excluded from the calculation of correlation with the regulatory measurements.

The sentence in line 163 seems not in place and would need more explanation and examples, how high does RH? show examples of how much data had to be removed due to this fact.

The sentence about negative values will be moved higher in the manuscript to where equation 1 is presented. The total of all of the QC steps removes about 10% of the available datain Part I. This information will be added to the ms.

Line 156-167 - Would be important to add information based on the data quality sections on how much data had to be removed due to each screening criteria (is case = event?). would have been efficient to show more examples not just one for dust and one for smoke. Show additional figures, and provide information in the text about the different examples so we will understand how Tables 2-3 were created.

We have added the slopes to the caption for Figure 2 and more text explaining how Tables 2+3 were generated. Given the current length of the paper, it is not clear why additional figures showing essentially the same thing will be useful We have modified Tables 2 and 3 to incorporate both the Barkjohn 2021 correction and the newer EPA correction to the PA data. The discussion has been expanded so it should be clear how these tables were generated.

Line 175- map needed for a non-American reader. Line 180 – nearby- how close, map, and distance will be helpful for the reader.

According to the coordinates given in each data for the Keeler data the PAS and regulatory sensors are within 30 meters of each other. The regulatory measurements are made with a Thermo Fischer model 1400a TEOM with a Thermo Fischer model 8500C conditioning system. Other information is given in Table S2.

Line 192 would be good to know when PAS and location are used or in this example, information could be added to table S1 so the reader will know what sensor was used for what event.

We have added these details to both figure captions along with regression parameters. The caption also notes that more details can be obtained in Table S1 and S2.

Line 211-216- the entire paragraph is unclear including fig 2 how were these calculated made, how many events were used to correlate the number of cases, did only a period of the event (e.g., smoke) was used, or sometimes data of period before or after as shown in Fig 1 were used. The R2 values are not significant so can you clarify what statistically significant are you talking about?

This paragraph has been rewritten to improve clarity. Figure 2 has been changed to emphasize the differences between the Barkjohn 2021 correction and the newer 5 part EPA correction to the PA data.

Line 230- what 17 events? You said there are 50 cases, can you indicate maybe in table S1 which one had both PM sizes that were used for this analysis?

Out of these 50 cases (part I) only 17 have $PM_{10}$ data. This line has been edited for greater clarity and the information has been added to Table S1.

Why not use the known threshold which is commonly used (PM2.5/PM10) for this comparison?

Figure 3 does not make any assumptions about the CAF. It only uses the data from the 17 events (out of 50) that had both $PM_{2.5}$ and $PM_{10}$. In fact, using the regulatory data for dust cases, the figure shows an average CAF of 0.64, or a fine aerosol fraction of 0.36.

Line 246- did you confirm the events were dust based on observation of meteorology conditions, and satellite? PM2.5/PM10 ratios?

Unlike the June 2020 dust events (which were observed over a large fraction of the U.S.), the Keeler dust periods are from a local dust source (the dry bed of Owens Lake) over different days between 2019-2022. Thus satellite is not going to be useful for this. Instead, we use the co-located measurements of $PM_{2.5}$ and $PM_{10}$.

Please add more information, and add dates of these events so the reader may be able to understand what you have done for this part. I think this is one of the most interesting parts of your paper but it seems you put the least amount of effort into it.

We have expanded the discussion here to provide more details. The dates of the data used are given in both the methods section and Table S2.

Line 266-267 what are the units for 150-250 and 190 are these nm or µg m-3?

This is a ratio of counts in the 0.3 µm bin to the counts in the 5µm bin. It is unitless. We have edited this line for better clarity.

Lines 285-289- show us these findings of the use of the new eq on the dust event of part one, this is a very important finding from this paper, and it seems the author barely provides information here. Also, would be nice to see exploring these sections a bit more, and show us similar to fig 1 what the dust events look like before and after using both corrections was this one dust event or several, and when did they occur, how long they lasted?

We have added a new table to the SI (Table S4) which shows the performance of the dust equation on the 50 events in Part I. We note that equation works reasonably well for the Keeler cases, but does not work as well for some of the events in Part I. As such this equation should be considered as prototype correction for dust, with more work needed.

Minor comments Line 54 and 55 - PM2.5 the 2.5 should be lowercase (PM2.5), there might be more of these that I missed Table 2-P value based on which statistical test?

Corrections made. Thank you. Table 2 has been significantly changed so this comment is no longer relevant. We note some comparisons of means in the text and P-values were calculated from standard t-tests.

Fig 4- units of the axis are missing

This is a histogram, so frequency is the correct label.

Reference used Sugimoto, N., Shimizu, A., Matsui, I., and Nishikawa, M.: A method for estimating the fraction of mineral dust in particulate matter using PM2:5-to-PM10 ratios, Particuology, 28, 114–120, https://doi.org/10.1016/j.partic.2015.09.005, 2016. Tong, D. Q., Dan, M., Wang, T., and Lee, P.: Long-term dust climatology in the western United States reconstructed from routine aerosol ground monitoring, Atmos. Chem. Phys., 12, 51895205, https://doi.org/10.5194/acp-12-5189-2012, 2012 Ardon-Dryer, K., Kelley, M.C., Xueting, X., and Dryer, Y.: The Aerosol Research Observation Station (AEROS), Atmos. Meas. Tech., https://doi.org/10.5194/amt-2021-270, 2022.

References added.

**Reviewer #2:**

I appreciate this manuscript because it's important that the air quality research community and users of low-cost air quality monitors, like the PurpleAir, understand that these devices can underestimate airborne dust concentrations by a lot.

My biggest concern is that the authors do not use the most recent U.S. EPA PurpleAir correction equation. Equation 1 (on Line 91) is the U.S.-wide correction equation that was published by Barkjohn et al. in Atmospheric Measurement Techniques (DOI: 10.5194/amt-14-4617-2021) but, since the time of that publication, the EPA has updated their PurpleAir correction approach because (a) Equation 1 underestimated very high smoke concentrations (as the authors note here) and (b) limitations associated with the PurpleAir API made a correction equation based on the PMS5003-reported PM$_{2.5}$ ATM concentrations more appealing. The updated EPA correction is a piecewise equation that (a) uses the PM$_{2.5}$ ATM value as a predictor (instead of the PM$_{2.5}$ CF=1 value) and (b) predicts high concentrations of wildfire smoke more accurately. See additional information in this presentation: https://cfpub.epa.gov/si/si_public_record_report.cfm?dirEntryId=353088&Lab=CEMM. EPA indicates that they started using the correction shown on Slide 14 in Summer 2021. The authors should (a) confirm with EPA that they are still using the piecewise correction shown in Slide 14 in this presentation, (b) confirm with EPA whether the piecewise correction shown on Slide 14 is the "EPA" correction being used on the PurpleAir map, and then (c) re-do their analysis using whichever correction EPA is currently using on the PurpleAir map and the AirNow Fire and Smoke map.

Thank you for pointing this out. We have redone all figures and tables in Part I to incorporate both the Barkjohn 2021 correction and the new (unpublished) EPA correction using a 5-part piece wise correction with weighting. While this does change some of the conclusions in Part I, the new equation has little influence on the conclusions in Part II (Keeler dust). For Part 1, the main change is that while the new equation does improve performance at high concentrations, but adds a moderate positive bias in the range of about 100-250 µg m$^{-3}$. See new Figures 1+ 2 and new Figure S6 below.

Additional comments:

Methods, Part I: What geographic area and date range did the authors' analysis span? Did the authors look for all pollution events at all U.S. regulatory monitoring sites that fell within a specific date range and had PM$_{2.5}$ concentrations > 40 µg m$^{-3}$ for 3 consecutive hours? Were the regulatory sites with PM$_{2.5}$ > 40 µg m$^{-3}$ identified first and then the nearest PurpleAir monitors identified? Did the authors impose any limit on how close the PurpleAir monitor had to be to the regulatory monitor for data from the two to be compared? What range of correlation coefficients between regulatory and PurpleAir data would have been accepted? What software did the authors use to analyze their data? Were pollution events identified systematically (this would be preferred!) or manually?

Events were identified manually with the requirement that a clear pollution event could be identified. These analyses were conducted with a combination of IBM SPSS and Microsoft Excel. We have added information to the Part I methods to address these questions.

Methods, Part II: What type of regulatory monitor is being used at the Keeler, CA site? The EPA map indicates that it's a TEOM. Please note the regulatory monitor make and model in the text. How far from the regulatory monitor was the PurpleAir monitor?

The Keeler regulatory measurements are made using a Thermo Fischer model 1400a TEOM with a Thermo Fischer model 8500C conditioning system. According to the coordinates given in each data for the Keeler data the PAS and regulatory sensors are within 30 meters of each other. This information has been added to the Part II methods.

Lines 224-225: "These differences are statistically significant for urban versus dust, but not for smoke versus dust at the 95% confidence level." Are the authors sure that the "size distribution" data reported by sensor aren't influenced by the absolute PM concentrations reported by the sensor? What do the PAS 0.3 µm/5 µm count ratios look like when high concentrations of smoke are present? Could the approach proposed to correct data from dust events lead to some smoke events being mischaracterized as dust events?

Yes, this is possible. We have added a new table to the SI (Table S4) which shows the performance of the dust equation on the 50 events in Part I. Based on our evaluation of the events in Part I, we find that our proposed correction equation has some significant flaws in some smoke and dust events. We have added more discussion on these flaws. Nonetheless, we choose to leave the equation in the paper as a means to point towards a direction that might be useful with more evaluation and refinement.

Technical corrections:

Lines 73-74: I think there are some words missing from this sentence. I suggest revising as something like "The $PM_{2.5}/PM_{10}$ ratio for smoke is also similar to that for urban pollution: 0.55–0.75 (Xu et al., 2017)."

**Citation**: https://doi.org/10.5194/amt-2022-265-RC2

Thank you. Corrected.

**New figures for main paper and SI:**

[Figure]

**Figure S1:** **Comparison of the PAS data corrected with the new EPA correction and the raw PAS ATM data, both in µg m⁻³.**

[Figure]

**Figure S2:** **Comparison of the PAS data corrected with the new EPA correction and the PAS data corrected with the Barkjohn 2021 equation, both in µg m⁻³.**

[Figure]

[Figure]

New Figure 1: Time series of hourly regulatory and PAS data PM$_{2.5}$ (w/new EPA and Barkjohn 2021 correction algorithms) for two events, #44 (smoke, top) and #45 (dust, bottom). Time is in UTC. Note that for the dust event (#45), the two correction schemes give identical results. Details on the sites used and the linear correlations with the regulatory data is given in Tables S1 and S2.

[Figure]

**New Figure 2. Comparison of mean bias (PAS-regulatory) for the hourly data for smoke and urban pollution events in part I using the Barkjohn 2021 and new EPA correction scheme. Data are binned by regulatory PM$_{2.5}$ in 50 µg m$^{-3}$ bins, as shown on the X axis. The values above the red points are the number of hourly datapoints in each bin, which is the same for both the Barkjohn and new EPA corrected values. gthe error bars show one standard deviation within that bin.**

[Figure]

**New Figure S6: Regulatory PM$_{2.5}$ versus PAS with new EPA correction at Keeler, CA for hours with regulatory PM$_{2.5}$ >25 µg m$^{-3}$. The data are separated by the CAF, as measured by the regulatory data and the PAS data are corrected using the new EPA algorithm.**

**New Table 1.** Peak regulatory PM$_{2.5}$, slope and R$^2$ results from analysis of regulatory and PAS data, with Barkjohn 2021 correction and new EPA correction, for 50 individual pollution events. (N gives number of events of each type, SD is standard deviation

| | Peak reg PM$_{2.5}$ (µg m$^{-3}$ | Slope using Barkjohn 2021 | Correl coef using Barkjohn 2021 | Slope using new EPA corr | Correl coef using new EPA corr |
|---|---|---|---|---|---|
| **Urban-avg** | 85.15 | 1.00 | 0.88 | 0.95 | 0.88 |
| **N** | 16.00 | 16.00 | 16.00 | 16.00 | 16.00 |
| **Sd** | 56.69 | 0.11 | 0.03 | 0.15 | 0.03 |
| **Smoke-avg** | 280.32 | 0.99 | 0.93 | 0.88 | 0.92 |
| **N** | 28.00 | 28.00 | 28.00 | 28.00 | 28.00 |
| **Sd** | 226.28 | 0.18 | 0.05 | 0.13 | 0.05 |
| **Dust-avg** | 59.76 | 5.54 | 0.85 | 5.53 | 0.85 |
| **N** | 6.00 | 6.00 | 6.00 | 6.00 | 6.00 |
| **Sd** | 7.91 | 1.13 | 0.07 | 1.10 | 0.07 |

**New Table 2.** Relationship between hourly regulatory PM$_{2.5}$ and PAS PM$_{2.5}$ with Barkjohn 2021 correction. Data are included for all simultaneous measurements for the 50 events. (N gives number of hours of data of each type). Mean bias is calculated as **PAS-regulatory.**

| | Mean Reg PM$_{2.5}$ (µg m$^{-3}$) | Mean corrected PAS PM$_{2.5}$ (µg m$^{-3}$) | Slope for Reg versus PAS w/corr (R$^2$) | Intercept (µg m$^{-3}$) | RMSE (µg m$^{-3}$) | Mean bias (µg m$^{-3}$) |
|---|---|---|---|---|---|---|
| **Urban (N=966)** | 33.9 | 28.7 | 1.02 (0.793) | 4.60 | 10.9 | -5.2 |
| **Smoke (N=6536)** | 66.4 | 66.0 | 1.08 (0.866) | -4.68 | 36.0 | -0.4 |
| **Dust (N=240)** | 30.5 | 6.4 | 4.98 (0.661) | -1.09 | 27.9 | -24.1 |

**New Table 4.** Relationship between hourly regulatory PM$_{2.5}$ and PAS PM$_{2.5}$ with new EPA correction. Data are included for all simultaneous measurements for the 50 events. (N gives number of hours of data of each type). Mean bias is calculated as **PAS-regulatory.**

| | Mean Reg PM$_{2.5}$ (µg m$^{-3}$) | Mean PAS PM$_{2.5}$ w/corr (µg m$^{-3}$) | Slope for Reg versus PAS w/corr (R$^2$) | Intercept (µg m$^{-3}$) | RMSE (µg m$^{-3}$) | Mean bias (µg m$^{-3}$) |
|---|---|---|---|---|---|---|
| **Urban (N=966)** | 33.9 | 30.3 | 0.950 (0.744) | 4.90 | 11.1 | -3.6 |
| **Smoke (N=6536)** | 66.4 | 77.3 | 0.807 (0.858) | 5.56 | 43.2 | 11.0 |
| **Dust (N=240)** | 30.5 | 6.4 | 4.99 (0.664) | -1.22 | 27.9 | -24.1 |

**New Table S3.** Regulatory PM$_{2.5}$, mean bias and slopes for each event using three different correction schemes. Slopes are given as per equation 2, so that a slope less than 1 indicates a positive bias in the PA corrected data (PA data are higher than regulatory data) and a slope greater than 1 indicates a negative bias in the PA corrected data (PA data are lower than regulatory data).

| Case # | Mean regulatory PM$_{2.5}$ (µg m$^{-3}$) | Mean bias w/Barkjohn 2021 corr. (µg m$^{-3}$) | Mean bias w/new EPA corr. (µg m$^{-3}$) | Mean bias w/new dust corr. (µg m$^{-3}$) | Slope using Barkjohn 2021 corr | Slope using new EPA correction | Slope using new dust correction |
|---|---|---|---|---|---|---|---|
| 1 | 24.4 | -4.5 | -3.7 | -3.7 | 0.92 | 0.85 | 0.85 |
| 2 | 21.3 | -9.1 | -8.8 | -8.5 | 1.07 | 1.00 | 1.00 |
| 3 | 21.3 | -8.3 | -7.8 | -7.8 | 0.94 | 0.86 | 0.86 |
| 4 | 21.3 | -6.8 | -6.5 | -6.5 | 0.99 | 0.95 | 0.95 |
| 5 | 47.5 | -13.2 | -12.4 | -12.4 | 0.85 | 0.85 | 0.85 |
| 6 | 46.1 | -14.5 | -12.5 | -12.5 | 0.98 | 0.95 | 0.95 |
| 7 | 29.0 | -4.7 | -2.9 | -2.9 | 0.95 | 0.81 | 0.81 |
| 8 | 44.1 | -4.9 | -1.4 | -1.3 | 1.15 | 1.18 | 1.18 |
| 9 | 49.1 | -6.3 | -5.8 | -5.9 | 0.95 | 0.96 | 0.96 |
| 10 | 22.3 | 1.4 | 2.0 | 9.3 | 1.15 | 1.12 | 0.22 |
| 11 | 29.8 | -7.0 | -6.4 | -6.2 | 0.94 | 0.88 | 0.89 |
| 12 | 36.3 | -1.3 | 1.3 | 2.8 | 1.04 | 1.04 | 1.08 |
| 13 | 52.5 | -5.1 | -2.7 | -2.0 | 1.27 | 1.31 | 1.32 |
| 14 | 18.8 | -2.4 | 0.2 | 0.2 | 1.05 | 0.82 | 0.82 |
| 15 | 19.4 | -1.1 | 1.9 | 3.4 | 0.93 | 0.73 | 0.71 |
| 16 | 33.4 | 3.4 | 10.3 | 10.3 | 0.85 | 0.83 | 0.83 |
| 17 | 20.1 | 3.9 | 3.9 | 3.9 | 0.85 | 0.85 | 0.85 |
| 18 | 35.1 | 2.8 | 3.0 | 3.6 | 0.96 | 0.96 | 0.96 |
| 19 | 28.9 | 6.0 | 6.2 | 4.3 | 0.81 | 0.81 | 0.84 |
| 20 | 14.6 | 2.7 | 3.0 | 3.0 | 0.7 | 0.68 | 0.68 |
| 21 | 14.7 | 1.9 | 2.4 | 2.4 | 0.75 | 0.72 | 0.72 |
| 22 | 35.9 | -2.1 | -1.8 | -1.8 | 1.02 | 1.02 | 1.02 |
| 23 | 35.9 | -1.1 | -0.8 | -0.8 | 1.05 | 1.05 | 1.05 |
| 24 | 38.8 | -1.6 | -1.3 | -1.3 | 0.98 | 0.98 | 0.98 |
| 25 | 35.9 | 0.8 | 1.1 | 1.1 | 1.00 | 1.00 | 1.01 |

| Case # | Mean regulatory PM$_{2.5}$ (µg m$^{-3}$) | Mean bias w/Barkjohn 2021 corr. (µg m$^{-3}$) | Mean bias w/new EPA corr. (µg m$^{-3}$) | Mean bias w/new dust corr. (µg m$^{-3}$) | Slope using Barkjohn 2021 corr | Slope using new EPA correction | Slope using new dust correction |
|---|---|---|---|---|---|---|---|
| 26 | 34.1 | 4.1 | 4.4 | 4.4 | 0.92 | 0.92 | 0.92 |
| 27 | 34.0 | 2.4 | 2.7 | 2.7 | 0.95 | 0.95 | 0.95 |
| 28 | 35.2 | 3.0 | 3.7 | 3.7 | 0.93 | 0.94 | 0.94 |
| 29 | 44.6 | 5.4 | 5.8 | 6.5 | 0.82 | 0.83 | 0.84 |
| 30 | 37.7 | 3.1 | 3.7 | 3.7 | 0.6 | 0.61 | 0.61 |
| 31 | 42.7 | 4.9 | 6.7 | 6.7 | 0.84 | 0.75 | 0.75 |
| 32 | 102.2 | -1.8 | 9.4 | 9.4 | 1.13 | 0.92 | 0.92 |
| 33 | 102.2 | -10.1 | -3.0 | -3.0 | 1.22 | 1.04 | 1.04 |
| 34 | 102.2 | -0.6 | 11.9 | 11.9 | 1.07 | 0.84 | 0.84 |
| 35 | 102.2 | -10.1 | -3.8 | -3.8 | 1.24 | 1.05 | 1.05 |
| 36 | 102.2 | -5.0 | 3.2 | 3.2 | 1.15 | 0.95 | 0.95 |
| 37 | 140.6 | 20.0 | 41.8 | 41.8 | 1.07 | 0.78 | 0.78 |
| 38 | 189.3 | -42.3 | -4.3 | 672.8 | 1.3 | 0.98 | 0.20 |
| 39 | 210.9 | -8.9 | 49.3 | 967.2 | 1.08 | 0.75 | 0.14 |
| 40 | 204.8 | -3.3 | 53.1 | 1105.3 | 1.14 | 0.76 | 0.12 |
| 41 | 68.2 | -20.6 | -17.5 | 0.8 | 1.29 | 1.06 | 1.02 |
| 42 | 196.7 | -6.8 | 51.8 | ND* | 1.06 | 0.73 | ND* |
| 43 | 148.3 | -15.4 | -10.0 | -10.0 | 1.00 | 0.90 | 0.90 |
| 44 | 67.4 | 7.7 | 10.9 | 10.9 | 0.81 | 0.70 | 0.70 |
| 45 | 31.3 | -25.5 | -25.5 | -15.9 | 6.76 | 6.7 | 0.72 |
| 46 | 33.1 | -26.8 | -26.8 | -26.8 | 7.03 | 6.98 | 6.98 |
| 47 | 34.1 | -25.9 | -25.9 | -0.7 | 4.55 | 4.69 | 0.46 |
| 48 | 28.3 | -21.5 | -21.5 | -21.5 | 5.67 | 5.64 | 5.64 |
| 49 | 28.5 | -22.6 | -22.6 | -7.2 | 4.69 | 4.64 | 0.63 |
| 50 | 27.9 | -22.4 | -22.4 | -21.4 | 4.55 | 4.51 | 0.67 |

---

## Author Response (AR2)

**Thank you for your careful read of the manuscript. We have incorporated all of these suggestions into the final ms. Note that line numbers refer to the document with tracked changes.**

I have relatively minor comments on this version of the manuscript. My biggest comment is:

1. Can the authors please emphasize throughout, especially for readers who are less familiar with the performance of the PMS5003 and PurpleAir sensors, that the PAS typically underestimates dust concentrations even before the EPA corrections are applied? The issue lies with limitations of the PMS5003 sensor, which the authors discuss on lines 65-67, and not just the EPA corrections. The EPA corrections make the existing underestimation even worse because they're optimized for the more commonly-observed urban and wildfire smoke pollution events. Can the authors put the raw PurpleAir data back in Figure 1?
**Done. I have added the raw PA data back into Figure 1.**

Some specific examples of text that I think could be revised to address this concern:
a. Lines 216-217: The authors could revise "…, indicating that both correction equations are significantly under-estimating the true concentrations by a factor of 6 or more." as something like "The raw PAS PM2.5 CF=1 values, the raw PAS PM2.5 CF=Atm values, the PM2.5 concentrations calculated using the Barkjohn 2021 correction, and the PM2.5 concentrations calculated using the new EPA correction all significantly underestimate the true PM2.5 concentrations."
**I have added text along these lines (line 219).**

b. Lines 228-229: The authors could revise "but they generate a large negative (low) bias for dust events." as "but they exacerbate the negative (low) bias for dust events."
**I have added text along these lines (line 234).**

c. Lines 243-244: "We show above that the PAS data, for both corrections, are substantially under-reporting PM2.5 concentrations during dust events." This is true, but the raw PAS data are also substantially under-reporting PM2.5 concentrations during dust events, right?
**Yes, I have added the text here for greater clarity (line 249).**

d. Lines 282-283: "…but for this group the PAS Barkjohn 2021 correction significantly underestimates the regulatory concentrations."
**Not sure what is being suggested here.**

2. Line 46: There is an extra "(" before "map.purpleair.com"
**Corrected.**

3. Line 72: I suggest revising "although the exact procedure is not documented by PurpleAir" as "although the exact procedure is not documented by Plantower". The PM1, PM2.5, and PM10 concentrations are reported by the Plantower sensor itself and I doubt that PurpleAir has access to Plantower's algorithms.
**I have added Plantower to this sentence, as suggested.(line 73)**

4. Line 73: "A number of field and laboratory studies have found that the PMS5003 size distributions are not correct." I think this sentence would be easier for readers to understand if the phrasing was more precise. I suggest revising as "A number of field and laboratory studies have found that the particle number size distributions reported by the PMS5003 are not correct."
**Changed. (Line 75)**

5. Line 95: "…but biased high compared to regulatory PM2.5 measurements." I suggest rephrasing this as "…but are often biased high compared to regulatory PM2.5 measurements in the United States." I suggest adding this qualifier "often" because, as the authors show in their results, PAS measurements underestimate PM2.5 concentrations for windblown dust. I suggest adding "in the United States" (or "in North America") because most of these studies took place in the U.S. and

results might be different in other parts of the world where PM2.5 concentrations and compositions are different. For example, the PMS5003 sensors are reportedly calibrated using ambient aerosol in Beijing, and they might not underestimate reference PM2.5 measurements there.
**Changed. (Line 97-98).**

6. Line 203: I think there are some words missing from this sentence. I suggest revising as: "…which provides 1366 hours of data spanning a 3.3-year period."
**Changed. (Line 207)**

7. Lines 257-259: "For the PAS data, we use the raw values for PM2.5 and PM10, since there are no known correction algorithms for the PM10 data." Did the authors use the CF=1 PM2.5 and PM10 values or the CF=Atm PM2.5 and PM10 values for these calculations? Please specify in the text and in the captions for Figure 3and Figure S6.
**Yes, these are CF=1.  Changes made (Line 264 and in figure captions).**

8. It would be helpful if the authors could add 1:1 lines to the graphs shown in Figures 5, 6, S7, S8, and S9.

**Done.**

**Finally, please note that a few details on the Keeler site were updated with current information from Chris Howard, Great Basin Unified Air Pollution Control District (lines 201-205, highlighted)**